# Antipyretic Mechanism of Bai Hu Tang on LPS-Induced Fever in Rat: A Network Pharmacology and Metabolomics Analysis

**DOI:** 10.3390/ph18050610

**Published:** 2025-04-23

**Authors:** Ke Pei, Yuchen Wang, Wentao Guo, He Lin, Zhe Lin, Guangfu Lv

**Affiliations:** 1School Pharmaceutical Sciences, Changchun University of Chinese Medicine, Changchun 130117, China; 18686658006@163.com (K.P.); wangyc01@ccucm.edu.cn (Y.W.); guowt@ccucm.edu.cn (W.G.); linhe@ccucm.edu.cn (H.L.); 2Jilin Province Ginseng Science Research Institute, Changchun University of Chinese Medicine, Changchun 130117, China

**Keywords:** Bai-Hu-Tang, fever, network pharmacology, metabolomics

## Abstract

**Background**: Bai Hu Tang (BHT) is a classic antipyretic in traditional Chinese medicine, however, there is little scientific evidence on the mechanism and material basis of its antipyretic effect. **Methods**: In LPS-induced febrile rats, after administration of BHT at 42 g/kg for half an hour, body temperature was measured at hourly intervals for 9 consecutive hours. Then, serum levels of TNF-α, IL-1β, and IL-6, and serum and cerebrospinal fluid (CSF) levels of AVP, cAMP, PGE2, Ca and CRH, and the remaining sera were used for metabolomics. These were then combined with network pharmacology methodology to further analyse the antipyretic effect of BHT and then dock key targets with differential components. **Results**: Administration of BHT to LPS-induced febrile rats significantly reduced elevated body temperature, TNF-α, IL-1β and IL-6 levels, but serum and CSF levels of AVP, cAMP, PGE2, Ca^2+^ and CRH were significantly elevated compared to the control group. Network pharmacological analyses indicated that the putative functional targets of BHT were regulation of immune responses, associated protein binding and inflammatory responses, and fine-tuning of phosphatase binding and activation of signalling pathways such as MAPK, PI3K, AKT, NF-kB, cAMP and inflammatory pathways. Metabolomic analysis showed that the antipyretic effect of BHT and its mechanism are likely to be involved in fatty acid metabolism, bile acid metabolism and amino acid metabolism in the organism, with L-arginine, glycyrrhetinic acid and N-acetylpentraxine as the main differential metabolites that play a significant role in heat recovery. The results also showed better docking of glycyrrhetinic acid with TNF-α, IL-6R, PTGS2. **Conclusions**: BHT provides a valuable adjunct to traditional clinical antipyretics by improving body temperature and metabolism and reducing inflammation.

## 1. Introduction

Nowadays, fever is one of the most researched diseases in China and abroad. Fevers influence the immune system of homeothermic animals and humans [1]. It is a general phenomenon connected to flu, COVID-19, inflammation, and infection [2]. Clinically, in traditional medicine, fever reaction has been characterised as a prolonged rise in body temperature above the normal temperature range, and often treated with removing excess heat and fire from the body [3]. Although fever is a Western term, and since the symptoms and characteristics are almost the same, it is categorised as into epidemic febrile disease in traditional medicine, and treated with antipyretic prescription according to the principles of febrile disease treatment in TCM [4].

Traditional medicine has a history of 5000 years, making it one of the oldest and most famous treatments in the world. Formulas are one of the most fundamental elements of traditional medicine, and the Bai-Hu-Tang (BHT) formula has a long history of use. The formula contained in the treatise on febrile disease of Shang Han Lun consists of Liquorice, Anemarrhena Rhizome, Gypsum and Rice, and is widely used in clinical medicine practice for the treatment of certain diseases, mainly to reduce fever. Further clinical trials on antipyretic effects and comparative studies with other treatments have been conducted. BHT may inhibit LPS-induced immunoinflammatory damage, exert immunoprotective and anti-injury effects and, thus, play a role in the prevention of endotoxic fevers [3]. It also promotes antigen phagocytosis, degradation, and cross-presentation in the liver to clear pyrogens [5]. Previous studies have shown that BHT contains a variety of chemical constituents [6,7] (among which gypsum has been suggested to play an important role [8]). However, the material basis and mechanism of action of the antipyretic properties of BHT remain to be further investigated.

In this study, a fever model was prepared by subcutaneous injection of LPS. Clinically, herbal medicines are usually administered orally to patients; therefore, traditional clinical means were used in the preparation of BHT. In addition to better simulating the absorption of the drug in vivo, gastric administration was used. Based on in vivo experiments combined with network pharmacology and metabolomics, the antipyretic mechanism and possible antipyretic constituents of BHT were investigated; the research outcome of this study will scientifically suggest that the peripheral antipyretic mechanism of BHT may be partly due to the inhibition of the endogenous thermogen TNF-α, preventing LPS-induced fever, and provide pharmacological interpretations in the traditional use of BHT.

## 2. Results

### 2.1. Active Compounds and Targets of BHT and Targets of PMD

A total of 120 eligible active ingredients of BHT were obtained from the TCMSP and the BATMAN-TCM database; additionally, 2176 targets of the active ingredients were obtained. A total of 593 targets were screened after removing duplicates. Fever was retrieved from the GeneCards (relevance score > 5), OMIM, Drugbank, and DisGeNET databases. A total of 1059 relevant targets were obtained by taking the intersection; Venn diagrams were created to identify 34 common targets for BHT and fever (Figure 1A).

A total of 34 BHT antipyretic targets were imported into the STRING database for protein-protein interaction analysis. The PPI network has 145 nodes and 634 edges (Figure 1B). The 34 core targets were ranked using the MCODE function in Cytoscape, and the top ten were TNF, CASP1, PTGS2, IL6R, CCL4, HMOX1, IL1R2, IFNGR1, CSF3R, and IFNA21 (Figure 1C). Cytoscape was used to establish the BHT-ingredients-disease-target. Five key components, including quercetin, ursodeoxycholic acid, lignans, kaempferol and chenodeoxycholic acid, were selected from the network topology analysis (Appendix A).

### 2.2. GO and KEGG Analysis of BHT Antipyretic Targets

GO and KEGG enrichment analysis of BHT antipyretic targets using Metascape and KEGG tools (The significance level was set at *p* < 0.01). GO functional enrichment analysis yielded a total of 333 terms, of which 203, 122, and 28 were for biological processes (BP), molecular functions (MF) and cellular components (CC), respectively, and the top 10 terms with the lowest p-values in each group were selected for visualisation. BP categories showed inflammatory response, cellular response to lipopolysaccharide, positive regulation of TNF production, regulation of insulin secretion, positive regulation of NO biosynthetic process, immune response, calcium-mediated signalling, prostaglandin metabolic process, MAPK cascade, and response to fructose. Analysis of the MF terms showed that cytokine activity, hormone activity, protein binding, IL-1 receptor binding, identical protein binding, growth factor activity, transcription regulatory region sequence-specific DNA binding, serotonin binding, enzyme binding, and heme binding were enriched. The enriched CC categories included extracellular space, extracellular region, integral component of presynaptic membrane, caveola, membrane raft, presynaptic membrane, cell surface, integral component of plasma membrane, and perinuclear region of cytoplasm. These findings indicate that BHT has antipyretic effects at different levels and different pathways. The top 10 significantly enriched KEGG pathways were selected (Figure 1D). Enriched pathways included Inflammatory bowel disease, IL-17 signalling pathway, JAK-STAT signalling pathway, Neuroactive ligand-receptor interaction, NF-kB, TNF, HIF-1, cAMP, and MAPK signalling pathway. These findings showed that BHT has antipyretic effects through the synergistic effects of multiple components and multiple targets (Figure 1E).

### 2.3. Screening of DEGs (Differential Genes)

In the sequencing dataset, we obtained 1064 DEGs (52 up-regulated and 198 down-regulated) by difference analysis between normal and fever samples (Figure 2A) and selected the top 20 fever-related genes (FRGs) for heatmap display (Figure 2B). After taking the intersection of the DEGs with the 10 FRGs, we obtained 3 fever-related DEGs (Figure 2C), and the expression level of FDX1 was significantly higher in the fever group. Compared with the normal group, the expression levels of IL6R, TNF-α, and PTGS2 were significantly higher in the fever group (*p* < 0.01) (see Figure 2D). The dispersion estimates results showed that the difference in the expression of IL6R, TNF-α and PTGS2 between the normal group and the fever group was statistically significant (*p* < 0.01) (Figure 2E).

### 2.4. Effect of BHT on Body Temperature and Serum Levels of Inflammatory Factors in Model Rats

As shown in Figure 3A,B, the changes in body temperature and body water content of the rats in each group were plotted. After modelling, the body temperature of the model group increased significantly. After administration of the corresponding drugs, the body temperature of each group decreased; only the calcined gypsum group did not show significant changes in body temperature (Figure 3A). To investigate the effects of BHT on inflammatory responses in fever rats, serum levels of the inflammatory mediators TNF-α, IL-1β, IL-6 and PTGS2 were measured by ELSIA. The fever group had significantly increased serum levels of TNF-α, IL-1β, IL-6 and PTGS2 (*p* < 0.05, *p* < 0.01) compared to the control group. BHT administration and gypsum resulted in significantly suppressed levels of TNF-α, IL-1β, IL-6 and PTGS2 (Figure 3B–D).

### 2.5. Effect of BHT on Serum and CFS Levels of Ca^2+^, AVP, cAMP, PGE2 and CRH

Compared with the control group, the serum and CSF levels of PGE2, cAMP, CRH and AVP in the rats in the model group were significantly higher (*p* < 0.01), and the serum Ca^2+^ content was increased (*p* < 0.01), while the CSF Ca^2+^ content was slightly decreased (*p* < 0.01), and all of them were found to be improved after administration of the corresponding drugs, but the effect of BHT was significantly better than that of gypsum, indicating that the fever in the LPS-induced rat fever model is closely related to the above relevant factors (Figure 4).

### 2.6. Qualitative and Quantitative Metabolite and Quality Control (QC) Analyses

Samples were analysed through qualitative and quantitative metabolite analysis, sample quality control analysis, total sample principal component analysis (PCA), cluster analysis, total sample assessment and preliminary determination of sample metabolite species, and analytical methods used were stable and could ensure reliability of data analysis. Based on the local metabolic database, the metabolites of the samples were qualitatively and quantitatively analysed using mass spectrometry; the results of the sample QC analysis are shown in Figure 5A–D, Appendix A. The results show that the curve overlap of the total ion flux for metabolite detection is high, i.e., the retention time and peak intensity are consistent, indicating that the mass spectrometry has good signal stability when the same sample is detected at different times, and that the high stability of the instrument provides an important guarantee for the reproducibility and reliability of the data; by performing Principal Component Analysis on the total samples of the experiment (including the quality control samples), the PCA results showed a trend of metabolome separation between the groups, indicating that there were significant differences in metabolomes between the sample groups, and the differences were significant (*p* < 0.05), indicating that the data modelling was reasonable and could be used for PCA analysis of the samples, the correlation can observe the biological replication between samples within the group, the closer the Pearson correlation coefficient r is to 1, the stronger the correlation between two replicated samples, the structure of this experiment shows that the samples are better replicated and the results are reliably analysed.

### 2.7. Cluster Principal Component Analysis (PCA)

Multivariate statistical analysis was used to analyse the principal components of the grouped samples of test samples to compare differences and to observe the magnitude of variability between groups of differences and between samples within groups, and the results are shown in Figure 6A,B. Through the principal component analysis, we found that the degree of variation between the groups was larger, the separation between the groups was larger, the clustering within the groups was more obvious, and there was some difference in their variables. From the position of the PCA plot and the score plot, the model group and the BHT group, the model group and the gypsum group, the difference was larger and there was a very good clustering within the groups, suggesting that the LPS-induced febrile model of the rat was successfully prepared. Meanwhile, the endogenous metabolite changes were different between each administration group, which may be due to the intervention effect of the administration group. In addition, there are different metabolite groups or metabolic network changes. Its antipyretic effect may be related to the administration of the test drug to regulate the metabolic network of the organism (i.e., the fever rats). In the BHT group and the gypsum group, the intervention of the fever model rats is stronger (Figure 6C,D). Through the overall qualitative and quantitative analysis of all samples, quality control analysis and sample reproducibility correlation analysis, we detected a total of 482 related metabolites. The reproducibility of the samples was found to be good, the established detection method reliable, the instrument stable, and the samples can be analysed acutely by split analysis, which is guaranteed for subsequent experiments. The higher the correlation coefficient of intra-group samples relative to inter-group samples, the more reliable the differential metabolites obtained. The Pearson’s Correlation Coefficient r (Pearson’s Correlation Coefficient) was used as an assessment of biological replicate correlation; the closer the absolute value of r is to 1, the stronger the correlation between the two replicate samples (Figure 6E).

### 2.8. Differential Metabolite Screening

The samples were further analysed by partial least squares discriminant analysis (orthogonal projections to latent structures, OPLS-DA) to screen for differential variables. As shown in Table 1, Figure 7A–G and Appendix A, the model group was compared with each administration group by screening for differential metabolites. We screened for 52 differential metabolites when comparing the model group with the normal group; a total of 38 potential biomarkers were found when comparing the model group with the positive group, of which 19 were down-regulated and 19 were up-regulated. When comparing the model group to the control group, we screened 52 differential metabolites, of which 9 metabolites were down-regulated and 43 metabolites were up-regulated. In the comparison between the model group and the positive group, a total of 38 potential biomarkers were screened, of which 19 were down-regulated and 19 were up-regulated. In the comparison between the model group and the BHT group, a total of 46 potential biomarkers were screened, of which 29 were downregulated and 17 were upregulated. In the comparison between the model group and the gypsum group, a total of 32 differential metabolites were screened, of which 4 were down-regulated.

### 2.9. GO and KEGG Pathway Enrichment Analysis for Differential Metabolites

When the body is feverish, significant biological changes occur in the body, especially in the metabolic system, and various metabolic processes in the body are affected. In this experiment, fever was induced in rats by saccharomyces cerevisiae, and LPS, after acting on the body, immune cells that produce endogenous pyrogen are activated to produce and release endogenous pyrogen (EP), which could act on the thermoregulatory centre and cause the release of thermotropic mediators (CRH, cAMP, PGE2, AVP, INF and Na^+^/Ca^2+^), and then change the thermoregulatory point and affect the body’s metabolism. The analysis identified 16 potential biomarkers related to LPS-induced fever in rats, of which 13 were upregulated and 3 were downregulated, involving arachidonic acid metabolism, regulation of TRP channels by inflammatory mediators, phenylalanine metabolism, d-arginine and d-ornithine metabolism, and bile acid metabolism. The main pathways involved were mTOR signalling pathway, the JAK-STAT signalling pathway, the NF-kB signalling pathway, and the HIF-1 signalling pathway. Significantly different metabolic processes and pathways were found in the febrile rats when compared to the untreated rats; these are shown in Figure 8A–D.

After network pharmacology and metabolomics analyses, the significantly different component, glycyrrhetinic acid, was obtained and docked to five key targets, which were found to be better docked to TNF-α, IL6R, PTGS2, and, thus, may be a potential target of action for the antipyretic effect of BHT (Table 2, Figure 9).

### 2.10. Effect of BHT on Brain Expression of TNF-α, IL6R, PTGS2 in Fever Rats

The protein expression of TNF-α, IL6R, and PTGS2 in the brain was significantly upregulated in the model group compared with the control group (*p* < 0.01); compared with the model group, each protein was significantly down-regulated in the BHT group (*p* < 0.05), and TNF-α and IL6R proteins were down-regulated in the gypsum group and the aspirin group (*p* < 0.05). There was no change in PTGS2 protein, and the calcined gypsum group had an effect only on TNF-α protein (*p* < 0.05) (see Figure 10).

## 3. Discussion

Although metabolic changes during fever have been reported previously, metabolomics and network pharmacology analyses have revealed different biomarkers under different conditions [9,10]. With the continuous advancement of modern medical technology, metabolomics tools are becoming more sophisticated and can be used for disease diagnosis and TCM research [11,12]. While network pharmacology has been used to identify differential expression of genes, the combination of the two can reveal intrinsic changes in an organism. However, there are many antipyretic drugs that have been in use for more than a decade. The underlying mechanisms of their antipyretic effects need to be further explored. In this study, metabolomics combined with network pharmacology provided a comprehensive understanding of the antipyretic mechanism of action of BHT.

Network Pharmacology is a discipline that uses computer technology and network analysis methods to study drug action, pharmacology, and drug interactions, providing a systematic approach to the study and understanding of drug combinations and mechanisms of action in TCM. It involves molecular-level research on formulas using existing data and drug network targets and potential interactions of formulas [13,14], followed by validation through molecular docking. Studies have reported that most fevers are caused by the action of intrinsic heat sources, which are good targets for antipyretic effects [15,16]. Several signalling pathways have been identified to exert antipyretic effects, such as the MAPK and NF-kB pathways. Li reported that Huanglian detoxification decoction exerts antipyretic effects by regulating the MAPK signalling pathway, thereby lowering body temperature [17]. Ye reported that gypsum has antipyretic potential through the NF-kB signalling pathway [8]. This study suggests that BHT may exert an antipyretic effect through multiple targets, pathways, and biological processes. This study will be followed by further validation through in vivo experiments in LPS-induced fever rat models.

After the success of the fever rat model, the body temperature increased over time, and the body temperature of the rat also decreased after administration of BHT. In the context of TCM, inflammation is thought to be a Yang-overload state, meaning that the balance between “Yin” and “Yang” is disturbed and leads to a febrile state. Pathogenic microorganisms invade the organism and activate immune cells to produce endogenous pyrogens (e.g., IL-6, IL-1β and TNF-α, etc.), which enter the bloodstream and affect the hypothalamic thermoregulatory centres, modulating the synthesis and release of mediators, shifting the thermoregulatory point upwards and inducing a febrile response, and molecular docking results showed that glycyrrhetinic acid was more closely related to TNF-α, IL6R, PTGS2. These three target proteins may be somewhat associated with antipyretic activity. Among these thermogenic sources, IL-6 is a cytokine with redundant and pleiotropic activity, whose synthesis is tightly regulated by transcriptional and post-transcriptional mechanisms that promote host defence by stimulating acute phase response, haematopoiesis, and immune response [18]; IL-1β is able to influence the adaptive and innate immune response, affecting T-cell maturation and B-cell proliferation, in addition to promoting inflammatory molecules (e.g., nitric oxide, phospholipase A2, cyclooxygenase-2 and prostaglandin E2) [19]; and TNF-α, a peptide produced by mononuclear phagocytes, is a significant mediator of the inflammatory process and clinical manifestations of acute infectious diseases [20]. PTGS2 (COX-2) is an enzyme involved in the synthesis of a class of biologically active substances called prostaglandins. When infection or tissue damage occurs in the body, the activity of PTGS2 increases, resulting in the synthesis of more prostaglandins, particularly PGE2, which causes vasodilation, stimulates nerve endings and raises body temperature through the action of the hypothalamic thermoregulatory centre, This leads to fever, which can be reduced by inhibiting the activity of PTGS2 and reducing the synthesis of PGE2, which in turn reduces inflammation and lowers body temperature, thus inhibiting PTGS2 can play an “antipyretic” role [21,22]. The results of this experiment showed that the expression levels of thermogenic factors (IL-6, IL-1β and TNF-α) in the serum of LPS-induced febrile model rats were significantly increased, conversely, the drug administration team was able to significantly inhibit the synthesis and release of these thermogenic factors [23], indicating that some drug components in BHT have an inhibitory effect on the secretion of some endogenous thermogens.

Selected PCA, OPLS-DA, and differential metabolite KEGG enriched with 17 possible biomarkers were obtained by examining serum metabolite changes in BHT-intervened rats through LPS-induced fever models in rats. Its antipyretic effects involve the biosynthetic pathways of amino acid metabolism, lipid metabolism, arachidonic acid metabolism, bile acid metabolism, arginine biosynthesis, and secondary metabolite biosynthesis. By analysing the synthesis and metabolic pathways of potential biomarkers, searching for proteins involved in their synthesis or metabolism and their signalling pathways, and combining them with the mechanism of fever, we found that seven of the 21 metabolites, namely L-O-phosphoserine, L-arginine, D-glucuronolactone, glycochenobalamin, prostaglandin E2, L-glutamic acid, and 18-hydroxycorticosterone, were highly correlated with fever.

By comparing the differences between the model groups and each other, examining specific changes in metabolites in the serum of each group after intervention in fever model rats, and combining the combined analysis with the mechanism of fever, it was found that 7 differential metabolites among the 21 regressed metabolites (L-O-phospho-serine, L-arginine, D-glucuronolactone, glycylcholic acid, PEG2, L-glutamic acid, 18-hydroxycorticosteroids) were highly correlated with fever, and the metabolic pathways in which they are involved are closely related to the regulation of TRP channels by inflammatory mediators (the NF-κB pathway, the HIF-1 pathway, the mTOR pathway, and the MAPK pathway) and the difference in metabolism between each group after drug administration is likely due to the different pharmacological basis of the tested drugs and the different metabolic networks and pathways by which they act in the organism.

In the comparison between the model group and the BHT group, we found eight significant differences in metabolites, among which L-arginine, D-glucuronolactone, glycyrhizinic acid, fenugreek alkaloids, N-acetylphenylalanine and L-O-phosphatidylserine showed up-regulation, whereas reduced glutathione and N-acetylpententrone showed down-regulation, which are involved in amino acid metabolism, lipid metabolism, biosynthesis of secondary metabolites and antibiotic-related pathways. L-arginine is a precursor for the synthesis of arginine vasopressin (AVP), which plays a significant role in normal thermoregulation and antipyrexia; over-regulation of L-arginine may result in increased AVP synthesis, limiting fever, promoting antipyrexia, and enhancing the suppression of stress-induced hyperthermia [24]. However, AVP can only partially block or reduce the lipopolysaccharide (LPS)-induced fever response [25], whereas prostaglandin E2 (PGE2) is an important biologically active substance produced and released in vivo mainly by stimuli such as inflammatory responses, infection, and tissue injury [26,27,28]; PGE2 can affect thermoregulation and fever through a variety of pathways, including inhibition of thermoreceptor activity [29], which can lead the thermoregulatory centre to mistakenly believe that body temperature has decreased and cause fever [30]. In inflammation and infection, increased production of PGE2 can inhibit vasoconstriction and reduce body surface heat dissipation, which, in turn, causes fever [31,32]. Therefore, elevated levels of PGE2 can cause fever. Some studies have shown that AVP can reduce the level of PGE2 in the blood [33,34]. At the same time, the down-regulation of the level of the PGE2 metabolite in the serum may further reduce the level of intrinsic thermogenic sources, thus exerting an antipyretic effect and regulating the hypothalamic thermoregulatory centre.

cAMP is an important cellular messenger molecule involved in the regulation of a variety of physiological processes, including cellular metabolism, cell proliferation, opening and closing of ion channels, etc. [35]. In the kidney, AVP can stimulate adenylate cyclase activity via the V2 receptor, which, in turn, leads to increased production of cAMP. The increased production of cAMP promotes the production of PGE2 [35], which, in turn, affects thermoregulation and fever, and also promotes the reabsorption of water and sodium, thus affecting fluid balance, disrupting the Na^+^/Ca^2+^ ratio inherent in the thermoregulatory centre, and Ca^2+^ efflux [36], which increases the Na^+^/Ca^2+^ ratio and causes an increase in body temperature [37,38]. The serum calcium ion concentration increases significantly after BHT is ingested and digested by the body, and Ca^2+^ enters the hypothalamus across the blood-brain barrier to reduce the Na^+^/Ca^2+^ ratio, producing a cooling effect. At the same time, serum cAMP levels decreased, reducing the production of PGE2 and increasing the cooling effect. This may be due to the fact that Ca^2+^ can regulate cAMP metabolism by inhibiting the activity of adenylate cyclase, which reduces the production of cAMP and, thus, PGE2, thereby exerting a cooling effect.

Glycyrrhetinic acid is a naturally occurring compound with a variety of pharmacological properties, including antioxidant, anti-inflammatory and antiviral effects. Several studies have shown that glycyrrhetinic acid may affect thermoregulation by modulating the inflammatory response and the immune system [39,40]. When the body produces inflammatory mediators and cytokines under conditions such as inflammation and infection, these substances can affect the thermoregulatory centre and lead to fever [41]; when pathogenic microorganisms invade the organism and activate immune cells to produce endogenous pyrogens (e.g., When pathogenic microorganisms invade the body and activate the immune cells to produce endogenous pyrogens, they act on the hypothalamic thermoregulatory centre after entering the bloodstream to regulate the synthesis and release of mediators and then move the thermoregulatory point upwards, causing a febrile response. In addition, it may also affect the activity of the hypothalamic thermoregulatory centre by influencing the nervous system and endocrine system, thus affecting the regulation of body temperature and the process of fever. However, the specific mechanism of glycyrrhetinic acid’s effect on thermoregulation and fever and its value for clinical application still need to be further researched and explored.

Comparison of the model with the raw gypsum group showed a regression of N-acetylpentraxin. The serum level of 5-HT increased significantly in rats after administration of raw gypsum. In vivo, pentazocine is first catalysed by serotonin methyltransferase, plus an acetyl group, to form N-acetyl-5-hydroxytryptamine, and it is then further catalysed by serotonin N-acetyltransferase to melatonin. This process modulates the metabolic levels of N-acetyl-pentazocine, also known as melatonin, which is a product of the conversion of pentazocine by a series of enzymes and is known for its antioxidant and anti-inflammatory properties [42]. Studies have shown that, during fever, the secretion of melatonin is affected, and there may be a decrease in melatonin levels, which can affect the function of the thermoregulatory centre and, thus, the regulation of body temperature [43,44]. In addition, melatonin is able to regulate the function of the immune system and promote the reduction of inflammation, thus contributing to the treatment of fever [45]. Meanwhile, 5-HT can influence the activity of the hypothalamic thermoregulatory centre by binding to 5-hydroxytryptamine receptors, thereby regulating body temperature [46]. It may also affect the febrile process by regulating the activity of immune cells and influencing the release of inflammatory mediators. However, further studies are needed to improve our understanding of the specific mechanisms of action of both.

Integrated network pharmacology and metabolomics modelling suggest that mTOR and cAMP pathways play important roles in the mechanism of fever relief, and we further speculate that BHT may act through these two pathways. However, experiments are needed to further illustrate that metabolism refers to life-sustaining chemical reactions, and their dysregulation has been implicated in the development of a wide range of diseases, such as fever. In this study, significant differential metabolites, including L-arginine, glycyrrhizinic acid and N-acetylpentetreotide, are strongly associated with fever and all play important roles in the process of fever resolution.

## 4. Materials and Methods

### 4.1. Liquid Preparation

All herbs containing BHT and calcined gypsum were commercially sourced from First Affiliated Hospital of Changchun University of Traditional Chinese Medicine Pharmacy, Changchun, China and identified by the Department of Traditional Chinese Medicine Resources of Changchun University of Traditional Chinese Medicine. The four medicines are all Taoist herbs. These included Liquorice (No. 230501), Anemarrhena Rhizome (230301), and Gypsum (23082001). For this study, we weighed 84 g of gypsum, added 8 times the amount of water, soaked for 30 min, and then decocted it by bringing it to a boil for 20 min, and adding four layers of gauze filtration filtrate (concentrated to 20 mL). Calcined gypsum, made from raw gypsum (23082001) (weighing 84 g, and produced through the same method as the gypsum) and Rice (230602), were deposited at the Changchun University of Chinese Medicine, China. Then, all herbs were mixed to 467 g (Conversion based on the amount of BHT in Shang Han Lun) and were extracted in boiling water (2 L) for 2 h. Gypsum was first boiled in water for 1 h. Then, other herbs were added and boiled for 1 h, respectively. The juice was filtered through three layers of gauze, concentrated to 100 mL, autoclaved at 121 °C for 15 min, and then stored in an airtight container at 4 °C until use.

### 4.2. Chemicals and Reagents

In this study, the following chemicals and reagent were used: LPS (from Wuhan PhD Biologicals Co., Wuhan, China): 0.01g, dissolved in 100 mL saline, prepared at a concentration of 0.1 g-L-1, ready-to-use); aspirin effervescent tablets (from AstraZeneca Pharmaceutical Co., Ltd. London, UK): 2 g, gastro-resistant, dissolved in 100 mL saline, and prepared at a concentration of 20 g/L), ABL80 portable multiparameter blood gas analyser (from Radiometer, Copenhagen, Denmark), OMRON infrared frontal thermometer (MC-872), OMRON electronic thermometer (MC-246), laboratory animal body composition analyser (from ImpediMed, Pinkenba, Australia), 5810R centrifuge (from Eppendorf, Hamburg, Germany), F50 enzyme labelling instrument (from Tecan, Männedorf, Switzerland), SLI-700 constant temperature chamber incubator, ultra performance liquid chromatography (UPLC), tandem mass spectrometry, MS/MS (Agilent 6495), rat tumour necrosis factor (TNF-α), interleukin-1β (IL-1β), interleukin-6 (IL-6), argipressin (AVP), phosphoadenosine (cAMP), prostaglandin E2 (PGE2), calcium (Ca), corticotropin-releasing hormone (CRH), and an ELSIA kit (from Jiangsu Enzyme Immunoassay Industry Co., Ltd., Nanjing, China).

### 4.3. Animals and Treatments

SPF male SD rats (250 ± 20 g) were commercially procured from Liao Ning Chang Sheng Experimental Animal Technology Co., Ltd. [Ben Xi China; license number: SCXK (Liaoning) 2020–0001] and maintained at standard temperature and humidity under a 12-h light/dark cycle. All animal experiments were approved by the Care and Welfare of Laboratory Animals Committee of the Institute of Changchun University of Chinese Medicine (2023227). The rats were divided into six groups randomly. Group I served as control, and received only physiological saline; group II was fever model, received only LPS (100 µg/kg) by intravenous injection; group III was gavaged with aspirin, group IV was gavaged with BHT (42 g/kg), group V was gavaged with gypsum, group VI was gavaged with calcined gypsum at the same time of LPS intravenous injection.

### 4.4. Monitor Body Temperature

One day before the experiment, the rats were shaved at the back of the neck with barber’s scissors, the skin was exposed, and an infrared frontal thermometer was used to detect the body temperature by placing the thermometer on the exposed skin. The temperature was recorded three times as the mean value of the basal body temperature before modelling, and the temperature was recorded every 1 h after modelling, and the mean value of the three measurements made by the electronic thermometer was recorded, and the temperature curves of the different groups were plotted by monitoring the temperatures for eight consecutive hours.

### 4.5. Acquisition of Blood and Tissue Samples

At 2 h after injection of LPS and BHT, fully conscious rats were restrained on a rat table, after bending the rat’s head backwards, when the hand felt the depression between the occipital ramus and the first cervical vertebra (the foramen magnum of the occipital bone), a disposable 5-gauge needle syringe was inserted in the midline of this depression at an angle of 45 degrees from the rat’s back and the needle was punctured into the medullary pool of the cerebellum, with a slight dull sensation to indicate the correct position, and then the cerebrospinal fluid (CFS) was aspirated. Then, 6 mL of blood was collected from the heart by puncture with a disposable blood collection tube. After blood collection, 5 mL blood was placed in sterile tubes and centrifuged (4 °C, 3000 rpm, 10 min) to obtain serum, stored at −80 °C.

### 4.6. Network Pharmacology Analysis

TCMSP (format: 19 November 2023, https://www.tcmsp-e.com/) and BATMAN (format: 22 November 2023, http://bionet.ncpsb.org.cn/batman-tcm/) were used to explore the main active compounds and targets of BHT. Oral bioavailability (OB) ≥ 30% and drug-like activity (DL) ≥ 0.18 were used to screen active compounds in the TCMSP database [47]. A score cutoff of >20 and *p* value < 0.05 was used to predict targets and active compounds in the BATMAN database. Fever-related targets were filtered from the GeneCards database (format: 22 November 2023, https://www.genecards.org/), OMIM database (format: 22 November 2023, https://www.ncbi.nlm.nih.gov/omim) and DisGeNET (format: 22 November 2023, https://www.disgenet.org/) [48]. Plotting Venn diagrams with BHT-related targets and fever-related targets to obtain intersections, and the common targets were obtained. Common targets were uploaded to the STRING database (format: 22 November, https://string-db.org/) to map protein-protein interaction networks. Cytoscape (v.3.7.0) tools are used to analyse BHT-component-target-fever networks and screen for core compounds and targets. Additionally, core targets uploaded to the Metascape database (format: 25 November 2023, https://metascape.org/) were used for GO and KEGG analyses.

### 4.7. GEO Sample Acquisition and Processing

Samples were retrieved from the GEO database (format: 27 November 2023, http://www.ncbi.nlm.nih.gov.hnucm.opac.vip/geo/) using the keyword ‘fever’ and restricting the data type (expression analysis by array) and organism (Homo sapiens) to obtain gene expression and clinical data. Gene expression and clinical data were retrieved from the GEO database, the gene symbols were annotated, and the data were corrected using Perl code. The expression levels of the BHT-treated fever core genes were obtained from each sample of the normal group and the fever group, and the sequencing data of the fever samples were obtained from GSE133751 (3 normal samples and 3 fever samples), and the core genes were entered into the GEO database for statistical analysis. Statistical analysis.

### 4.8. Metabolomic Analysis

The sample was thawed on ice and three volumes of ice-cold methanol were added to 1 volume of plasma/serum. The mixture was whirled for 3 min and centrifuged it at 12,000 rpm at 4 °C for 10 min. Then, we collect the supernatant and centrifuge it at 12,000 rpm at 4 °C for 5 min. Finally, we collected the supernatant again to be used for LC-MS/MS analysis.

The sample extracts were analysed using an LC-ESI-MS/MS system (UPLC, Shim-pack UFLC SHIMADZU CBM A system, https://www.shimadzu.com/; MS, QTRAP^®^ System, https://sciex.com/). The analytical conditions for UPLC were as follows: Waters ACQUITY UPLC HSS T3 C18 column (1.8 µm, 2.1 mm × 100 mm) with a column temperature of 40 °C, a flow rate of 0.4 mL/min, an injection volume of 2 μL, a solvent system of water (0.04% acetic acid):acetonitrile (0.04% acetic acid), and a gradient program of 95:5 V/V at 0 min, 5:95 V/V at 11.0 min, 5:95 V/V at 12.0 min, 95:5 V/V at 12.1 min, and 95:5 V/V at 14.0 min.

LIT and triple quadrupole (QQQ) scans were acquired on a triple quadrupole-linear ion trap mass spectrometer (QTRAP), QTRAP^®^ LC-MS/MS System, equipped with an ESI Turbo Ion-Spray interface, operating in positive and negative ion mode and controlled by Analyst 1.6.3 software (Sciex). The ESI source operation parameters were as follows: source temperature 500 °C; ion spray voltage (IS) 5500 V (positive), −4500 V (negative); ion source gas I (GSI), gas II (GSII), curtain gas (CUR) were set at 55, 60, and 25.0 psi, respectively; the collision gas (CAD) was high. Instrument tuning and mass calibration were performed with 10 and 100 μmol/L polypropylene glycol solutions in QQQ and LIT modes, respectively. A specific set of MRM transitions was monitored for each period, according to the metabolites eluted within this period.

### 4.9. Molecular Docking

3D structures of target proteins, including TNF-α, CASP1, PTGS2, IL6R, CCL4 and HMOX1 were obtained from the Protein Data Bank (PDB) and the common major components obtained from network pharmacology and metabolomics were retrieved from the PubChem database, saved as mol2 format files, converted to PDB format files, and then saved as pdbqt files and exported to the working directory after hydrogenation and dehydrogenation of small molecules by the AutoDock 1.5.6 tools. Similarly, the receptor proteins were saved as a pdbqt format file and then molecularly docked to the pre-processed proteins and ligands by Vina using the semi-flexible docking method and finally Discovery Studio 4.5 was used to visualise the 2D plots.

### 4.10. Detection of Biochemical Indicators

CFS and serum were collected from rats and processed according to the kit instructions, the absorbance was measured at 450 nm on an enzyme meter and the sample concentration was calculated according to the linear regression equation of the standard curve to determine the levels of Ca^2+^, AVP, cAMP, PGE2, CRH in serum, and CFS, and the levels of TNF-α, IL-1β, IL-6, and PTGS2 in serum of rats.

### 4.11. Western Blot

Brain samples were taken from −80 °C refrigerator, and tissue lysate and protease inhibitor were added. The tissue was broken by ultrasonication, then centrifuged at 4 °C, 12,000× *g* for 20 min. The supernatant was aspirated, and the protein concentration was determined by BCA protein assay kit, and the protein samples were denatured and processed, and then made into protein samples, which were subjected to uploading, electrophoresis and membrane transfer, and then added with 5% skimmed milk powder and sealed by shaking the bed at room temperature for 2 h. Samples were incubated at 4 °C overnight. TNF-α, IL6, and a PTGS2 primary antibody were added and incubated overnight at 4 °C. After washing with TBST, secondary antibody (1:5000) was added and incubated for 2 h at room temperature. The samples were analysed using the Chemiluminescence Analysis System.

### 4.12. Statistical Evaluation

Data are expressed as mean ± standard deviation (S.D.). A T-test was used to compare the level of significance (P) between control and administered drugs, in which statistical significance was considered as *p* < 0.05. SPSS 13.0 software was used for data processing.

## 5. Conclusions

The current study integrated network pharmacology and comprehensive targeted metabolomics technology to investigate the antipyretic effect of BHT. The results showed that the body temperature of rats in the BHT treatment group decreased, affecting thermoregulatory factors and exerting a favourable antipyretic effect. BHT plays an antipyretic role by regulating amino acid metabolism, tryptamine secretion, and terpene metabolism. In addition, it is hypothesised that BHT may inhibit the increase in body temperature by activating the explicit mTOR and cAMP signalling pathways. The potential clinical applications of the antipyretic effects of BHT should be considered in the future.

## Figures and Tables

**Figure 1 pharmaceuticals-18-00610-f001:**
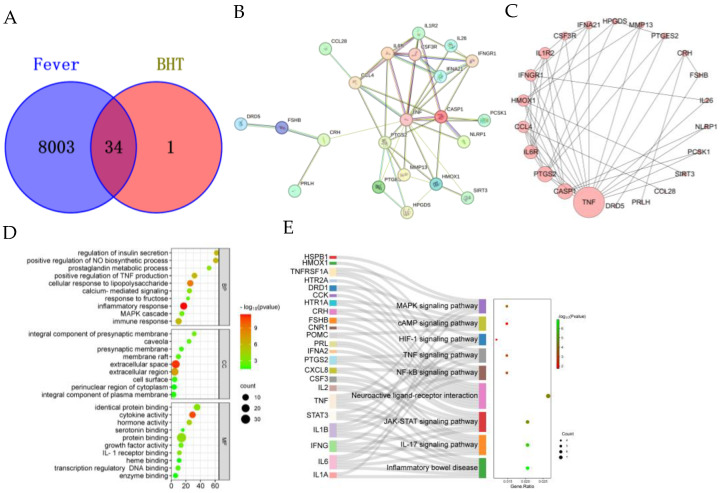
Network pharmacological analysis. (**A**). Target intersection Venn diagram. (**B**). Protein-protein interaction (PPI) network. (**C**). Network Topology Analysis. The shape size represents the degree value size. (**D**). Gene Ontology (GO) function analysis. The different colour symbols represent targets (blue), disease (green), signalling pathway (yellow), and compounds (pink). (**E**). Pathway Enrichment Analysis. Performed using Kyoto Encyclopedia of Genes and Genomes (KEGG).

**Figure 2 pharmaceuticals-18-00610-f002:**
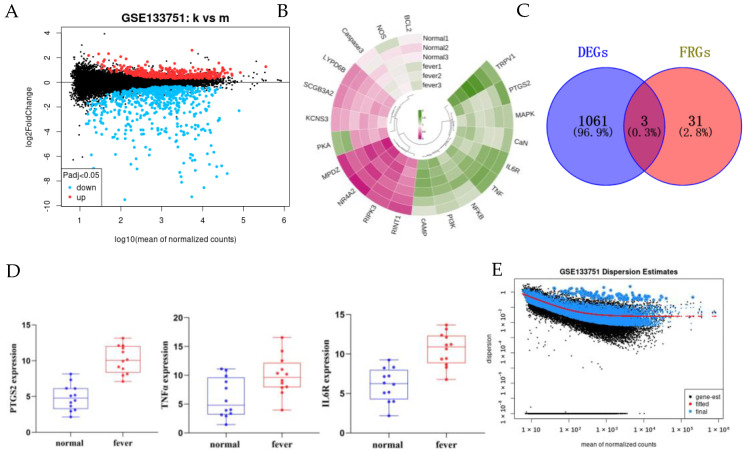
GEO Analytics related data graphs. (**A**). DEGs volcano map(Black dots represent unchanged genes). (**B**). DEGs Heat Map. (**C**). Intersection of DEGs and FEGs. (**D**). Histogram of IL6R, TNF-α and PTGS2 expression. (**E**). DEGs Dispersion Estimates Volcano Map.

**Figure 3 pharmaceuticals-18-00610-f003:**
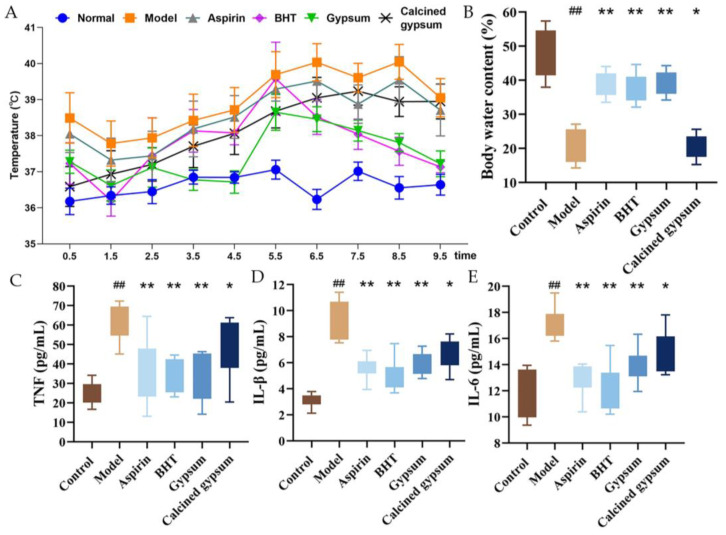
Effects of BHT on body temperature and TNF-α, IL-1β and IL-6 levels in the serum of LPS-induced fever rats. (**A**) body temperature, (**B**) Body water content, (**C**) TNF-α, (**D**) IL-1β, (**E**) IL-6 levels. Data are shown as mean ± SD for n = 8. * *p* < 0.05 and ** *p* < 0.01 versus model, ^##^
*p* < 0.01 versus the control.

**Figure 4 pharmaceuticals-18-00610-f004:**
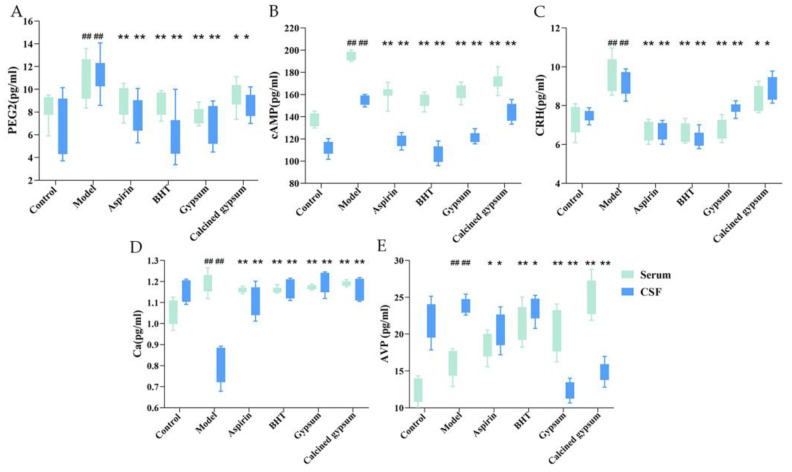
Effects of BHT on Ca^2+^, AVP, cAMP, PGE2, CRH levels in the serum, and CFS of LPS-induced fever rats. (**A**) PGE2, (**B**) cAMP, (**C**) CRH, (**D**) Ca^2+^, (**E**) AVP levels. Data are shown as mean ± S.D for n = 8. * *p* < 0.05 and ** *p* < 0.01 versus model, ^##^
*p* < 0.01 versus the control.

**Figure 5 pharmaceuticals-18-00610-f005:**
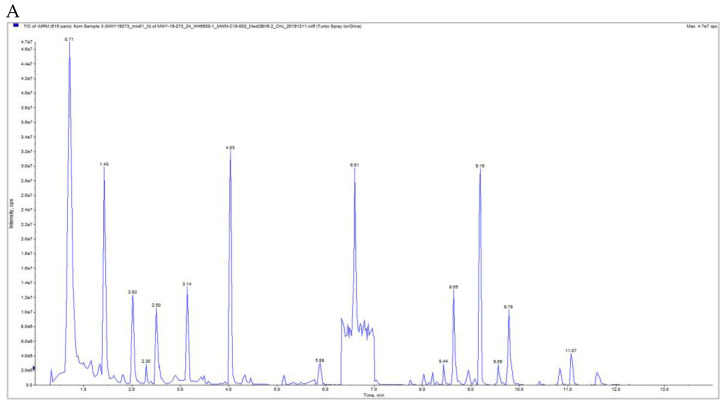
Qualitative and quantitative metabolite analysis. (**A**,**B**) Mixed Sample Mass Spectrometry TIC Diagram, (**C**,**D**) MRM Metabolite Assay Multi-Peak Plot.

**Figure 6 pharmaceuticals-18-00610-f006:**
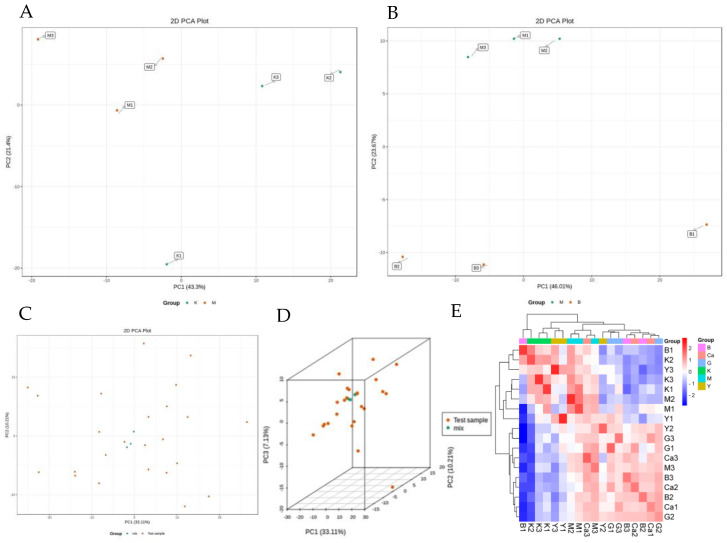
Cluster PCA. (**A**) Plot of PCA for K and M, (**B**) Plot of PCA for G and B, (**C**) Plot of PCA scores for each group of samples against mass spectral data of QC samples, (**D**) 3D plot of PCA, (**E**) Correlation plot between samples, Note: the horizontal coordinate indicates the name of the sample, the vertical coordinate indicates the name of the corresponding sample, and the colour represents the magnitude of the correlation coefficient value. Note: K (normal group), M (model group), B (BHT group), G (Gypsum group).

**Figure 7 pharmaceuticals-18-00610-f007:**
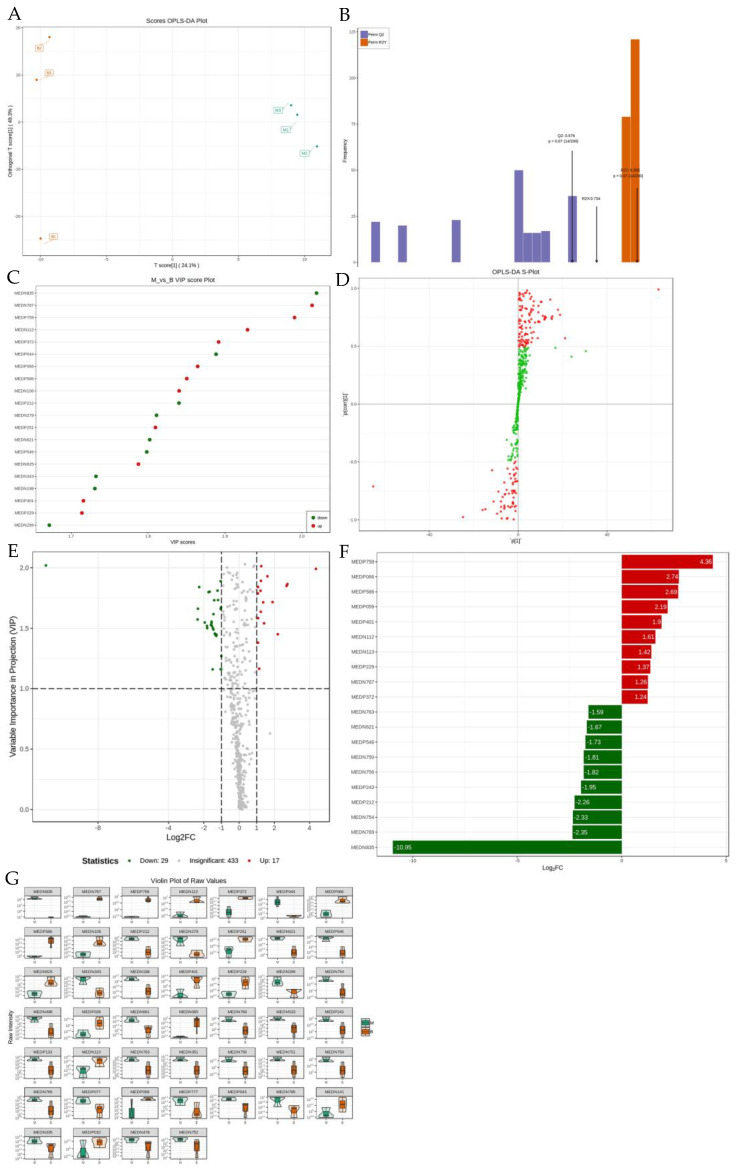
Cluster PCA. (**A**) OPLS-DA score plot for G and B, (**B**) OPLS-DA validation plot for G and B, (**C**) differential metabolite VIP value plots for G and B, (**D**) OPLS-DA S-plot for G and B, (**E**) differential metabolite volcano plots for G and B, (**F**) histogram of multiplicity of differences between G and B, and, (**G**) differential metabolite violin plots for G and B.

**Figure 8 pharmaceuticals-18-00610-f008:**
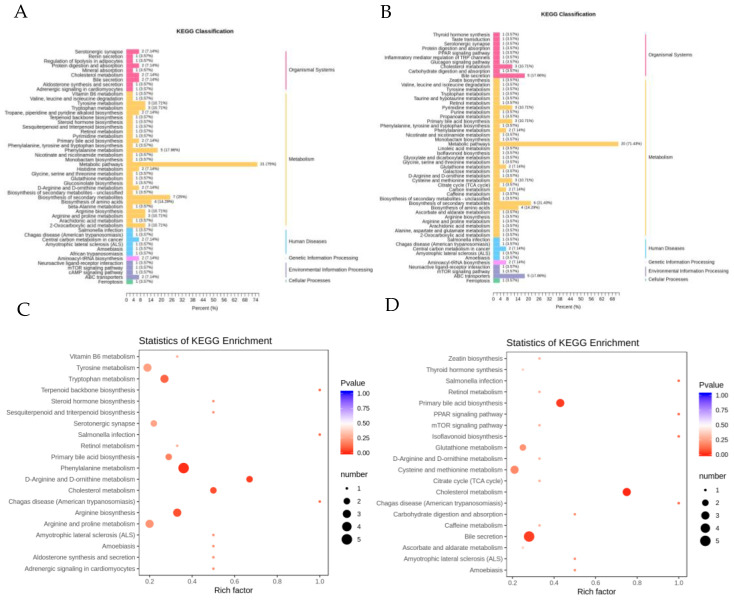
Cluster PCA. (**A**) Classification maps for K and M differential metabolic GO enrichment analysis, (**B**) Classification maps for G and B differential metabolic GO enrichment analysis. Note: The vertical coordinate is the name of the GO enrichment process, and the horizontal coordinate is the number of metabolites annotated for that process and the ratio of their number to the total number of metabolites for which they were annotated. (**C**) KEGG pathway enrichment map for K and M differential metabolites, (**D**) KEGG pathway enrichment map for G and B differential metabolites. Note: The horizontal coordinate indicates the rich character corresponding to each pathway, the vertical coordinate is the pathway name, and the colour of the point is the p-value, with redder indicating more significant enrichment. The size of the dot represents the number of different metabolites enriched.

**Figure 9 pharmaceuticals-18-00610-f009:**
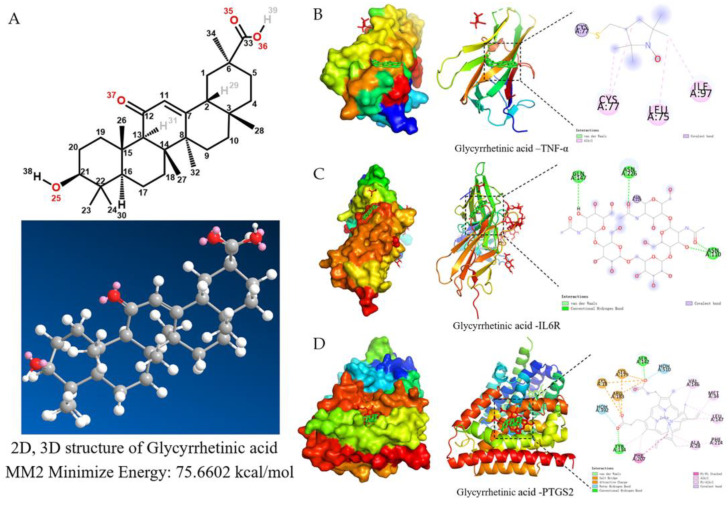
Diagram of molecular docking patterns. (**A**) Chemical structure of Glycyrrhetinic acid, (**B**) Glycyrrhetinic acid and TNF-α, (**C**) Glycyrrhetinic acid and IL6, (**D**) Glycyrrhetinic acid and PTGS2.

**Figure 10 pharmaceuticals-18-00610-f010:**
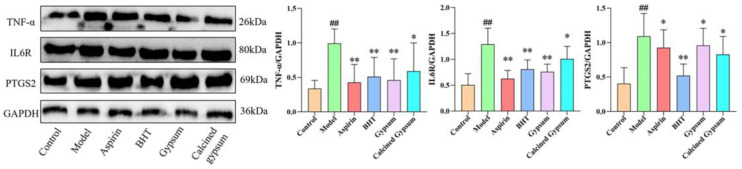
Effect of BHT on protein reporter of TNFα, IL6R and PTGS2. Associated protein bands and histograms. * *p* < 0.05 and ** *p* < 0.01 versus model, ^##^ *p* < 0.01 versus the control.

**Table 1 pharmaceuticals-18-00610-t001:** K vs. M and M vs. B difference metabolite tables (sorted by VIP value).

K vs. M		M vs. B	
Index	Compounds	Index	Compounds
MEND019	Glycocholic Acid	MEND044	Diethyl Malonate
MEND043	L-Theanine	MEND066	D-glucuronolactone
MEND061	Tryptophan Betaine	MEND106	Uridine 5-Monophosphate
MEND097	Glycocholic Acid	MEND112	Glycyrrhetinic Acid
MEND112	Glycyrrhetinic Acid	MEND198	Phenylacetyl-L-Glutamine
MEND113	L-arginine	MEND212	Fenugreek Alkaloids
MEND176	N-Phenylacetylglycine	MEND229	N-acetylphenylalanine
MEND324	19Z-docosahexaenoic Acid	MEND251	L-O-phosphatidylserine
MEND334	Xanthosine	MEND279	Glutathione (Reduced Form)
MEND338	4-Oxoretinol	MEND299	N-acetylpententrone
MEND356	Glycyrrhizic Acid	MEND343	Oxidised Lipid
MEND394	14Z-eicosatetraenoic Acid	MEND372	Epinephrine
MEND416	17Z-eicosapentaenoic Acid	MEND401	2′-Deoxyuridine
MEND440	8Z,14Z-eicosatrienoic Acid	MEND546	Phenols and its Derivatives
MEND454	10E,14Z-eicosatetraenoic Acid	MEND586	Uridine 5-Monophosphate
MEND509	Methylmalonic Acid	MEND624	Organic Acid and its Derivatives
MEND523	Aminomalonic Acid	MEND625	Trigonelline
MEND531	Glycochenodeoxycholic Acid	MEND759	Lactose
MEND611	Cortisol 21-Acetate	MEND767	L-Theanine
MEND720	Tryptophan Betaine	MEND835	Amino Acid Metabolomics

**Table 2 pharmaceuticals-18-00610-t002:** Molecular Docking Binding Mode Summary Table.

Ligand Name	Receptor	Receptor Residue	Type of Interaction	Binding Energy (kcal/mol)
Glycyrrhetinic acid	IL6R	GLN147 (A chain)	Conventional Hydrogen Bond	−9.79
Glycyrrhetinic acid	IL6R	ASN226 (A chain)	Conventional Hydrogen Bond	−8
Glycyrrhetinic acid	IL6R	ASN110 (A chain)	Conventional Hydrogen Bond	−7.91
Glycyrrhetinic acid	TNF-α	CYS77 (A chain)	Covalent bond	−6.99
Glycyrrhetinic acid	TNF-α	LEU75 (A chain)	Covalent bond	−8.81
Glycyrrhetinic acid	TNF-α	ILE97 (A chain)	Covalent bond	−7.25
Glycyrrhetinic acid	PTGS2	LYS28 (A chain)	Salt Bridge	−9.21
Glycyrrhetinic acid	PTGS2	LYS179 (A chain)	Salt Bridge	−8.26
Glycyrrhetinic acid	PTGS2	ARG183 (A chain)	Attractive charge	−7.59
Glycyrrhetinic acid	PTGS2	SER142 (A chain)	Conventional Hydrogen Bond	−7.99
Glycyrrhetinic acid	PTGS2	HOH392 (A chain)	Water Hydrogen bond	−6.91
Glycyrrhetinic acid	PTGS2	HOH444 (A chain)	Water Hydrogen bond	−5.86
Glycyrrhetinic acid	PTGS2	HOH510 (A chain)	Water Hydrogen bond	−8.36
Glycyrrhetinic acid	PTGS2	PHE214 (A chain)	Pi Alkyl	−10.59
Glycyrrhetinic acid	PTGS2	PHE207 (A chain)	Pi-Pi Stacked	−12.12
Glycyrrhetinic acid	PTGS2	MET34 (A chain)	Alkyl	−10.05
Glycyrrhetinic acid	PTGS2	VAL146 (A chain)	Alkyl	−8.02
Glycyrrhetinic acid	PTGS2	TYR134 (A chain)	Conventional Hydrogen Bond	−7.32
Glycyrrhetinic acid	PTGS2	ALA28 (A chain)	Pi Alkyl	−5.98
Glycyrrhetinic acid	PTGS2	LEU147 (A chain)	Alkyl	−5.23

## Data Availability

Data is contained within the article.

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
