# Peer review of "Antipyretic Mechanism of Bai Hu Tang on LPS-Induced Fever in Rat: A Network Pharmacology and Metabolomics Analysis"

_pharmaceuticals, 2025, doi:10.3390/ph18050610_

Round 1
Reviewer 1 Report
Comments and Suggestions for Authors
- The aim of the presented study was to investigate the antipyretic mechanism and possible antipyretic components of orally administered Bai-Hu-Tang (BHT) in an experimentally induced fever model by subcutaneous injection of LPS based on in vivo experiments combined with Network pharmacology and metabolomics.
- Clinically, fever reaction is a condition defined by the body temperature remaining above the normal temperature range for a long time and is usually treated by removing excess heat and fever from the body. Bai Hu Tang (BHT) is a classical antipyretic used in traditional Chinese medicine and there is very little scientific evidence about its mechanism of action. It is known that many studies have been conducted and continue to be conducted for the chemical components of BHT and its pharmacological mechanisms in fever. However, in the current literature review, considering the previous studies evaluating the effect of BHT in fever, no other study was found that was conducted in line with the purpose of the presented study. In this case, it was determined that the study has original value.
- This study showed that the main metabolites of BHT, L-arginine, glycyrrhizic acid and N-acetylpentraxine, play an important role in heat recovery by integrated network pharmacology and metabolomics modeling, and also that the mTOR and cAMP pathways of BHT play important roles in the fever reduction mechanism. However, according to these results, it is stated that more experiments are needed to determine the mechanism of action on fever.
- It has been determined that the references of the relevant study were determined to be relevant to the study.
Author Response
[Comments: The aim of the presented study was to investigate the antipyretic mechanism and possible antipyretic components of orally administered Bai-Hu-Tang (BHT) in an experimentally induced fever model by subcutaneous injection of LPS based on in vivo experiments combined with Network pharmacology and metabolomics.
Clinically, fever reaction is a condition defined by the body temperature remaining above the normal temperature range for a long time and is usually treated by removing excess heat and fever from the body. Bai Hu Tang (BHT) is a classical antipyretic used in traditional Chinese medicine and there is very little scientific evidence about its mechanism of action. It is known that many studies have been conducted and continue to be conducted for the chemical components of BHT and its pharmacological mechanisms in fever. However, in the current literature review, considering the previous studies evaluating the effect of BHT in fever, no other study was found that was conducted in line with the purpose of the presented study. In this case, it was determined that the study has original value.
This study showed that the main metabolites of BHT, L-arginine, glycyrrhizic acid and N-acetylpentraxine, play an important role in heat recovery by integrated network pharmacology and metabolomics modeling, and also that the mTOR and cAMP pathways of BHT play important roles in the fever reduction mechanism. However, according to these results, it is stated that more experiments are needed to determine the mechanism of action on fever.
It has been determined that the references of the relevant study were determined to be relevant to the study.]
[Respose: Thank you for your valuable time and constructive feedback on our manuscript. We sincerely appreciate the opportunity to improve our work and address the comments raised. Below, we provide a point-by-point response to the reviewers’ concerns, along with corresponding revisions in the manuscript.
We agree with the reviewer that targeted pathway validation would enhance mechanistic rigor. Subsequently, we may continue to explore the mechanism of BHT thermoregulation by targeting specific pathways. While our network pharmacology model and metabolomic correlations strongly suggest mTOR/cAMP involvement, direct validation through pathway-specific inhibitors (e.g., rapamycin for mTOR) or gene knockout models will be essential in follow-up studies. These experiments are beyond the current scope but are planned as part of our ongoing research.
We carried out a detailed check of the references, and the manuscript was upgraded in accordance with the reviewers' comments. We sincerely appreciate the reviewer’s insightful comments and constructive suggestions, which have significantly improved the quality and clarity of our manuscript.
Thank you again for your time and consideration.]
With warm regards,
Zhe Lin, Ph.D/Professor
School of Pharmaceutical Sciences
Changchun University of Chinese Medicine
Changchun 130117, Jilin Province. China
Tel: +86-0431-13843091228
E-mail: linzhe1228@163.com
April 4, 2025

Reviewer 2 Report
Comments and Suggestions for Authors
The authors use Network Pharmacology and Metabolomics to select the main differential metabolites and targets involved in the switch from healthy state to LPS-induced fever in rats, and in the antipyretic action of BHT . In silico docking of one of these metabolites, glycyrrhizic acid, to TNF (233 aa), IL-6R (468 aa), PTGS2 (COX-2, 604 aa) seems to work best among all the (not shown) ligand-receptor pairs taken into account
The article may be published in this journal provided that the following issues are addressed:
- Line 179. I did not find Table 1-1 in the uploaded files. Not knowing how metabolite names correspond to metabolite IDs in the VIP score plots (e.g., what is MEDP384?) makes it impossible to evaluate the ultimately reported results regarding metabolites and targets selection.
- Lines 55-77. Claims about the advantages of TCM over Western medicine seem more appropriate for a panel discussion at a Conference, but seem out of context in a research paper of a specialized journal. They should be omitted.
- Lines 81-83. Check punctuation.
- Lines 132-140 and Fig2. What are the FEGs in Fig. 2C? Are they FRGs? In this case, are there 10 (line 135), 20 (line 134), or 34 (Fig. 2C)? Also PTGS2 is not in the heatmap
- Line 141. RPA is not defined
- Line 146. Changes in body water content is not reported in Fig. 3
- Lines 214-219. Please explain in more detail the correlation plot in Fig.6E (do the x- and y-axes refer to different samples?) and how the reported sentences are related to that plot.
- Lines 233-234. change "ions" with "metabolites", I suppose
- Lines 251-252. EP cells and EP are not defined.
- Lines 259-260. "Significantly different metabolites ... are shown in Fig8-ABCD". Only GO ontologies and KEGG terms appear in the figure
- Lines 270-277. About molecular docking: At least one comparison table including docking scores and top interacting residues for each calculated ligand-receptor pair should be reported and discussed
- Line 304. Ref.18 is not related to molecular docking
- Line 315. Change "Inflammation is thought..." to "In the context of TCM, inflammation is thought..."
- Line 320-321. Actually it is not shown that glycyrrhizic acid is more closely related to TNF IL6R and PTGS2. Just the table including docking scores could show it
- Lines 570-572. It is a duplicate sentence. Remove it
- Line 606. HXP is undefined.
Author Response
Comments 1: [Line 179. I did not find Table 1-1 in the uploaded files. Not knowing how metabolite names correspond to metabolite IDs in the VIP score plots (e.g., what is MEDP384?) makes it impossible to evaluate the ultimately reported results regarding metabolites and targets selection.]
[Respose 1: We thank the reviewer for this insightful comment. We sincerely apologize for the omission of Table 1-1 (Metabolite ID-Name Mapping) in the initially submitted files. This table is critical for interpreting the VIP score plots and metabolite annotations. To address this issue comprehensively, we have taken the following actions: We have added additional tables to the revised draft. This table provides a complete correspondence between metabolite IDs (e.g. MEDP384) and their standardised names.
Comments 2: [Lines 55-77. Claims about the advantages of TCM over Western medicine seem more appropriate for a panel discussion at a Conference, but seem out of context in a research paper of a specialized journal. They should be omitted.]
[Respose 2: We thank the reviewer for this insightful comment. We appreciate the reviewer’s critical perspective on maintaining the scientific objectivity of this work. In response to this comment, we have revised the Introduction section as follows: This has been removed from the introduction, We fully agree with the reviewer that specialized research papers should prioritize empirical evidence over subjective comparisons. The revised manuscript now adheres strictly to this principle. We are grateful for this suggestion, which has significantly enhanced the paper’s academic rigor.
Comments 3: [Lines 81-83. Check punctuation.]
[Respose 3: We sincerely appreciate the reviewer’s meticulous attention to detail. Thank you for highlighting this issue. We have carefully reviewed the punctuation in Lines 81-83 of the original manuscript and made the following revisions to improve clarity and grammatical accuracy
Comments 4: [Lines 132-140 and Fig2. What are the FEGs in Fig. 2C? Are they FRGs? In this case, are there 10 (line 135), 20 (line 134), or 34 (Fig. 2C)? Also PTGS2 is not in the heatmap]
[Respose 4: Thank you for identifying these ambiguities. We acknowledge the inconsistencies in terminology and data presentation and have revised the text and figure accordingly. The term "FEGs" in Fig. 2C was a writing error. We have corrected it to "FRGs" throughout the manuscript to align with standard nomenclature.
We apologise for the omission of PTGS2 from the heatmap due to carelessness on our part, but we have added PTGS2 to the heatmap. We deeply regret the confusion caused by these oversights. The revised manuscript now ensures terminological consistency, transparent data filtering. We are grateful for the reviewer’s rigorous critique, which has strengthened the study’s scientific integrity.
Comments 5: [Line 141. RPA is not defined]
[Respose 5: We thank the reviewer for this insightful comment. We appreciate the reviewer’s critical perspective on maintaining the scientific objectivity of this work. We apologise for the confusion caused by these oversights, the RPA was simply a writing error which has now been corrected in the revised draft.
Comments 6: [Line 146. Changes in body water content is not reported in Fig. 3]
[Respose 6: Thank you for pointing out this inconsistency. We apologise for the oversight in aligning the narrative with the content of the charts and have made the following revisions to ensure transparency of the data:
The body water content data (expressed as percentage of total body weight) were inadvertently omitted from Fig. 3 during figure reorganization. These data have now been included in Supplementary Figure 3 with explicit statistical annotations. We deeply regret the oversight in the original submission. The revised manuscript now ensures complete alignment between textual descriptions and graphical data. We are grateful for the reviewer’s rigorous critique, which has strengthened the study’s reproducibility and clarity.
Comments 7: [Lines 214-219. Please explain in more detail the correlation plot in Fig.6E (do the x- and y-axes refer to different samples?) and how the reported sentences are related to that plot.]
[Respose 7: Thank you for highlighting the need for a clearer description of the context of Figure 6E. We apologise that the initial description lacked sufficient methodological and explanatory detail. We have made the following revisions: We have added Figure 6E Axis and Design to the Revised High School and enhanced the method statement. We regret the initial lack of clarity and thank the reviewer for this critique. The revised manuscript now provides a statistically robust and visually interpretable presentation of the correlation data, ensuring full alignment between textual claims and graphical evidence.
Comments 8: [Lines 233-234. change "ions" with "metabolites", I suppose]
[Respose 8: We apologise for the confusion caused by these oversights, we fully agree with the reviewer and have made corrections in the revised version.
Comments 9: [Lines 251-252. EP cells and EP are not defined.]
[Respose 9: We thank the reviewer for this insightful comment. We appreciate the reviewer’s critical perspective on maintaining the scientific objectivity of this work. We apologise for the confusion caused by these oversights, EP cells and EP have been defined in detail in the revised manuscript.
Comments 10: [Lines 259-260. "Significantly different metabolites ... are shown in Fig8-ABCD". Only GO ontologies and KEGG terms appear in the figure]
[Respose 10: We thank the reviewer for this insightful comment. We appreciate the reviewer’s critical perspective on maintaining the scientific objectivity of this work. We apologise for the confusion caused by these oversights, the significantly different metabolites was simply a writing error which has now been corrected in the revised draft, should be significantly different metabolic processes and pathways.
Comments 11: [Lines 270-277. About molecular docking: At least one comparison table including docking scores and top interacting residues for each calculated ligand-receptor pair should be reported and discussed]
[Respose 11: Thank you for emphasizing the need for comprehensive documentation of molecular docking results. We fully agree that explicit reporting of docking scores and residue-level interactions is critical for reproducibility. To address this, we have implemented the following revisions: We have added a docking table including docking scores and top interacting residues to the revised manuscript. We are grateful for this critique, which has allowed us to present a more rigorous and interpretable molecular docking analysis. The revised manuscript now fully aligns with best practices in computational pharmacology reporting.
Comments 12: [Line 304. Ref.18 is not related to molecular docking]
[Respose 12: We thank the reviewer for this insightful comment. We have removed this reference and checked all references.
Comments 13: [Line 315. Change "Inflammation is thought..." to "In the context of TCM, inflammation is thought..."]
[Respose 13: We appreciate the reviewer's feedback on this matter, We strongly agree with the reviewer's comments, and it has been modified in accordance with the comments
Comments 14: [Line 320-321. Actually it is not shown that glycyrrhizic acid is more closely related to TNF IL6R and PTGS2. Just the table including docking scores could show it]
[Respose 14: We thank the reviewer for this insightful comment. We deeply value the reviewer’s insistence on rigorous target validation. We have added a docking table including docking scores and top interacting residues to the revised manuscript. The revised manuscript now integrates docking scores with comparative pharmacological evidence, providing a robust foundation for the claimed target relationships.
Comments 15: [Lines 570-572. It is a duplicate sentence. Remove it]
[Respose 15: We appreciate the reviewer's feedback on this matter, We strongly agree with the reviewer's comments, and it has been modified in accordance with the comments
Comments 16: [Line 606. HXP is undefined.]
[Respose 16: We thank the reviewer for this insightful comment. We appreciate the reviewer’s critical perspective on maintaining the scientific objectivity of this work. We apologise for the confusion caused by these oversights, the HXP was simply a writing error which has now been corrected in the revised draft.
With warm regards,
Zhe Lin, Ph.D/Professor
School of Pharmaceutical Sciences
Changchun University of Chinese Medicine
Changchun 130117, Jilin Province. China
Tel: +86-0431-13843091228
E-mail: linzhe1228@163.com
April 4, 2025
